

# Sequence/structural analysis of xylem proteome emphasizes pathogenesis-related proteins, chitinases and *β*-1, 3-glucanases as key players in grapevine defense against *Xylella fastidiosa*

Sandeep Chakraborty[1,*], Rafael Nascimento[1,2,*], Paulo A. Zaini[2,*], Hossein Gouran[1], Basuthkar J. Rao[3], Luiz R. Goulart[2,4] and Abhaya M. Dandekar[1]

[1] Department of Plant Sciences, University of California, Davis (UC Davis), CA, United States of America

[2] Institute of Genetics and Biochemistry, Federal University of Uberlândia, Campus Umuarama, Uberlândia Minas Gerais, Brazil

[3] Department of Biological Sciences, Tata Institute of Fundamental Research, Mumbai, Maharashtra, India

[4] Department of Medical Microbiology and Immunology, University of California, Davis (UC Davis), CA, United States of America

[*] These authors contributed equally to this work.

Corresponding author
Abhaya M. Dandekar,
amdandekar@ucdavis.edu

## ABSTRACT

**Background**. *Xylella fastidiosa*, the causative agent of various plant diseases including Pierce's disease in the US, and Citrus Variegated Chlorosis in Brazil, remains a continual source of concern and economic losses, especially since almost all commercial varieties are sensitive to this Gammaproteobacteria. Differential expression of proteins in infected tissue is an established methodology to identify key elements involved in plant defense pathways.

**Methods**. In the current work, we developed a methodology named CHURNER that emphasizes relevant protein functions from proteomic data, based on identification of proteins with similar structures that do not necessarily have sequence homology. Such clustering emphasizes protein functions which have multiple copies that are up/down-regulated, and highlights similar proteins which are differentially regulated. As a working example we present proteomic data enumerating differentially expressed proteins in xylem sap from grapevines that were infected with *X. fastidiosa*.

**Results**. Analysis of this data by CHURNER highlighted pathogenesis related PR-1 proteins, reinforcing this as the foremost protein function in xylem sap involved in the grapevine defense response to *X. fastidiosa*. *β*-1, 3-glucanase, which has both anti-microbial and anti-fungal activities, is also up-regulated. Simultaneously, chitinases are found to be both up and down-regulated by CHURNER, and thus the net gain of this protein function loses its significance in the defense response.

**Discussion**. We demonstrate how structural data can be incorporated in the pipeline of proteomic data analysis prior to making inferences on the importance of individual proteins to plant defense mechanisms. We expect CHURNER to be applicable to any proteomic data set.

## INTRODUCTION

*Xylella fastidiosa* (*X. fastidiosa*) is a xylem-limited pathogen associated with diseases in many economically important plants, including Pierce's Disease of grape (PD) and Citrus Variegated Chlorosis (CVC) (*Chatterjee, Almeida & Lindow, 2008*). *X. fastidiosa* lives within the host's water-conducting xylem vessels, where it forms biofilms believed to be responsible for reduced hydraulic conductance caused by clogging of the vessels, and not increased cavitation and embolism of xylem elements (*McElrone, Sherald & Forseth, 2003*).

The xylem is composed mainly of lignified vessels that are used for the transportation of water, mineral nutrients and metabolites throughout the vascular system, and in long-distance signaling in response to biotic and abiotic stresses (*De Bernonville et al., 2014*). Xylem sap contains small molecular weight inorganic compounds, organic substances (*Metzner et al., 2010*), amino acids and proteins (*Biles & Abeles, 1991*). Recent improvements in genomic and proteomic technologies are accelerating the characterization of these proteins. The xylem sap proteome has been characterized in different plants, which has been shown to contain several protein families such as metabolic enzymes, stress-related proteins and signal transduction proteins (*Buhtz et al., 2004*; *Dafoe & Constabel, 2009*; *Djordjevic et al., 2007*; *Kehr, Buhtz & Giavalisco, 2005*; *Ligat et al., 2011*; *Rep et al., 2002*; *Zhang et al., 2015b*). These include glycoside hydrolases, peroxidases, chitinases, lipid transfer proteins, proteases, lectins, pathogenesis-related proteins and cell wall structural proteins. The differential accumulation of proteins in xylem sap and apoplast fluid following pathogen infection has been investigated in some pathosystems, clearly indicating that protein composition changes during plant-pathogen interactions, both by the response of the host and by secreted effectors from the pathogen (*Floerl et al., 2008*; *Gawehns et al., 2015*; *Houterman et al., 2007*; *Pu et al., 2016*; *Rep et al., 2002*; *Subramanian et al., 2009*).

*Vitis vinifera* cv. Chardonnay xylem sap protein composition was previously analyzed by two-dimensional gel electrophoresis, which identified only ten proteins (*Agüero et al., 2008*). While the role played by xylem proteins in defense against biotic stress has been established in other plant species, the only information available about grapevine xylem sap proteins and their importance to plant response during *X. fastidiosa* pathogenesis came from the pioneering work and Yang and collaborators *(2011)* and a recent contribution by *Katam et al. (2015)*. While the former showed that thaumatin-like and heat-shock proteins were significantly overexpressed in PD-resistant varieties of grape (*Yang et al., 2011*), the latter found several uniquely expressed proteins ($\beta$-1, 3-glucanases, 10-deacetyl baccatin III-10-O-acetyl transferase-like, COP9, and aspartyl protease nepenthesin precursor proteins) in PD-tolerant muscadine grape (*Katam et al., 2015*).The phenolic compounds altered during this plant-pathogen interaction has also been investigated (*Wallis & Chen, 2012*), expanding our understanding of the host molecular response to *X. fastidiosa* infection.

Moreover, the comparison of the xylem sap proteome of PD-tolerant and PD-susceptible grapevine species revealed the presence of few proteins that might be directly involved with plant defense against *X. fastidiosa* (*Basha, Mazhar & Vasanthaiah, 2010*). These studies however rely on protein sequence-based approaches for peptide mapping and identification (*Altschul et al., 1997*; *Fenyo & Beavis, 2003*), which limits exploring the wealth of information generated in proteomic analysis. Proteins with no sequence homology often possess similar enzymatic capabilities due to convergent evolution (*Gherardini et al., 2007*) and promiscuity (*Chakraborty & Rao, 2012*; *Copley, 2003*; *Jensen, 1976*); two well-studied phenomena analyzed by considering structural features. As structural data analysis can focus on several properties of target proteins rather than the one-dimensional alignments inherent to sequence-based methods, a structure-based data analysis approach is not well established for proteomics. We present a simple method for classifying protein sets using metrics derived from protein fold which can suggest putative functions to uncharacterized proteins by structural similarity. Our pipeline also performs a more localized perspective and analyzes specific active site residues to determine functional equivalence (*Chakraborty et al., 2011*; *Kleywegt, 1999*). This approach was applied here to better understand the molecular basis of the interaction between this xylem-colonizing bacterium and grapevines, on data generated by comparing the composition of the xylem sap proteome of infected plants with that of healthy plants. Our analysis pipeline (CHURNER) was able to confirm previous studies cited above and identify novel proteins not previously detected or yet uncharacterized, and is freely available to be used with other proteomic data sets.

## MATERIALS & METHODS

### Xylem sap collection and protein precipitation

Xylem sap was collected from six 3-year-old grapevines (*Vitis vinifera* cv. 'Thompson Seedless') located at the University of California Davis (Armstrong field). Three of these plants were mechanically inoculated with *Xylella fastidiosa* Temecula1 12 months prior to sap collection. The presence of *X. fastidiosa* in the xylem sap of infected plants was confirmed using anti- *X. fastidiosa* antibodies in a Double Antibody Sandwich ELISA (Agdia, USA) following manufacturer's instructions (Fig. S1). Xylem sap (30–50 mL per plant) was collected overnight in the second week of spring by drip of the cut stem of *X. fastidiosa*-infected and non-infected plants. To initiate sap collection, an apical segment of approximately 10 cm was cut from the stem and the vine terminal introduced into a collection tube sealed with parafilm. Xylem sap was lyophilized followed by protein precipitation using TCA/Acetone (*Gorg et al., 2000*). The pellets were resuspended in 300 μL of PBS (pH 7.4) and total protein was quantified using BCA Protein Assay Kit (Thermo Fisher Scientific) following manufacturer's instructions for subsequent SDS-PAGE and LC-MS/MS analysis.

### Protein preparation, mass spectrometry analysis and NMR imaging

Proteins were precipitated using ProteoExtract[TM] Protein Precipitation kit (Calbiochem) followed by dehydration overnight in a sterile fume hood. The protein pellet was then resuspended in 50 mM AmBic (pH 8.0) and 100 μg subjected to an in-solution tryptic digestion. The digested peptides were analyzed using a QExactive mass spectrometer

(Thermo Fisher Scientific) coupled with an Easy-LC (Thermo Fisher Scientific) and a nanospray ionization source. One microgram of digested peptides were loaded onto a trap (100 micron, C18 100° A 5U) and desalted online before separation using a reverse phased column (75 micron, C18 200° A 3U). The gradient duration for separation of peptides was 60 min using 0.1% formic acid and 100% acetonitrile as solvents A and B, respectively. Raw data was analyzed using X!Tandem (*Fenyo & Beavis, 2003*) and visualized using Scaffold version 4.4.1 (Proteome Software, OR). Samples were searched against UniProt databases appended with the cRAP database, which recognizes common laboratory contaminants. Reverse decoy databases were also applied to the database prior to the X!Tandem searches. Peptide identifications were accepted if they could be established at greater than 95% probability by the Peptide Prophet algorithm (*Keller et al., 2002*; *Nesvizhskii et al., 2003*) with Scaffold delta-mass correction. Protein identifications were accepted if they could be established at greater than 99% probability and contained at least 2 identified peptides (see File S1 for raw data of identified proteins. All supplemental material is available at http://dx.doi.org/10.5281/zenodo.50672). Proteins that contained similar peptides and could not be differentiated based on MS/MS analysis alone were grouped to satisfy the principles of parsimony. For relative protein quantification of xylem sap from infected and non-infected plants, the QSpec statistical framework (Version 2, https://sourceforge.net/projects/qprot/) was used to assign significance to differentially regulated proteins, using a Bayes factor >10 (*Choi, Fermin & Nesvizhskii, 2008*).

Nuclear magnetic resonance imaging ($^1$H-MRI) was done in an Avance 400 spectrometer equipped with Bruker DRX console microimaging accessory according to *Dandekar et al. (2012)*. Stem transverse sections of all non-infected and infected plants were collected between internodes located at the top (apical), middle and bottom of the central stem (three cuts per plant) and subjected to MRI. Fig. 2 shows representative results.

## Downstream *In silico* methods

We have written custom programs to automate the extraction of protein sequences, their annotation through the BLAST command line (*Camacho, 2008*), obtaining homologous PDB structures, and getting pairwise structural homology (*Konc & Janezic, 2010*) from proteome data mined with Prophet/Scaffold program (*Keller et al., 2002*) (see example dataset in File S1). These programs were integrated in the CHURNER pipeline using freely available BioPerl (*Stajich et al., 2002*) modules and Emboss (*Rice, Longden & Bleasby, 2000*) tools (additional documentation and scripts available as Files S2 and S3). As manual steps, we performed gene ontology of the differentially expressed proteins using the statistical overrepresentation test from the PANTHER protein classification system (*Mi et al., 2013*), and FATCAT (*Ye & Godzik, 2004*) to supervise and validate the structural homology. All protein structures were rendered by the PyMol Molecular Graphics System, version 1.7.4 Schrödinger, LLC (http://www.pymol.org/). Detection of putative signal peptides and target sub-cellular locations of proteins were done with SignalP 3.0 (*Bendtsen et al., 2004*) and TargetP 1.1 (*Emanuelsson et al., 2007*), respectively. Protein sequences used in sequence and structural alignments were devoid of signal sequences to better represent mature proteins. Congruence of specific active site residues to determine functional equivalence

between proteins was performed with CLASP (*Chakraborty et al., 2011*). Adaptive Poisson–Boltzmann Solver (APBS) and PDB2PQR packages were used to calculate the electrostatic potentials of all the atoms in the protein (*Baker et al., 2001*; *Dolinsky et al., 2004*). The APBS parameters were set as described previously in *Chakraborty et al. (2011)*. APBS writes out the electrostatic potential in dimensionless units of kT/e where k is Boltzmann's constant, T is the temperature in K and e is the charge of an electron, used to calculate the pairwise potential differences.

## RESULTS & DISCUSSION

### The CHURNER workflow for proteome analysis

CHURNER implements tandem analysis of sequence and structure of proteins to highlight potential functional similarities not obvious from simple sequence alignments. It uses simple Perl scripts to obtain the pairwise sequence and structural homology scores from BLAST and ProBiS, respectively (Fig. 1). The structural homology is then checked with FATCAT by the FCTSIG significance test (*Ye & Godzik, 2004*). We used these well established methods with different algorithms to detect structural similarity. This helps us corroborate the results between them (PZ scores). FATCAT treats the protein as flexible, allowing twists in the reference protein (akin to a real protein) and minimizes the number of rigid-body movements for the best structural alignment. ProBiS relies on common surface structural patches rather than global conservation in finding structural similarity, with the reasoning that the surface residues are more critical since they determine ligand or protein-protein interactions. The source code, working directory and a README script for the current example is available as Supplemental Information (see Methods). As a working example we used proteins identified by LC-MS/MS from xylem sap of grapevines infected with *X. fastidiosa* as described below. By using CHURNER we were able to highlight the Pathogenesis-Related proteins (PR-1) as the main proteins of grapevine defense against *X. fastidiosa* to be secreted in xylem sap and also to unravel structural similarity between $\beta$-D-glucan exo-hydrolase and chitinases, which has no known reference in existing literature. To the best of our knowledge, this is the first attempt to analyze proteomic data for differentially expressed proteins based on structural features. A walkthrough of the steps implemented by CHURNER will be demonstrated next.

### Our working example: xylem sap proteins identification by LC-MS/MS

In this study, proteins from xylem sap of *X. fastidiosa*-infected and non-infected grapevines were lyophilized, precipitated by TCA/acetone and analyzed by SDS-PAGE (Fig. S2). A total of 91 proteins (Table 1) were identified herein by LC-MS/MS with at least two peptides sequenced per protein. SignalP and TargetP were used to predict the presence of signal peptides and sub-cellular localization, respectively, in all protein sequences. Signal peptides were found in 70 proteins (77%) of which 67 were predicted to be secreted while the others are directed towards an undetermined sub-cellular location.

### Differentially expressed proteins in xylem sap of infected plants

Previous reports investigating differentially expressed transcripts and proteins in xylem sap of *X. fastidiosa*-infected plants have provided a wealth of information regarding the plant
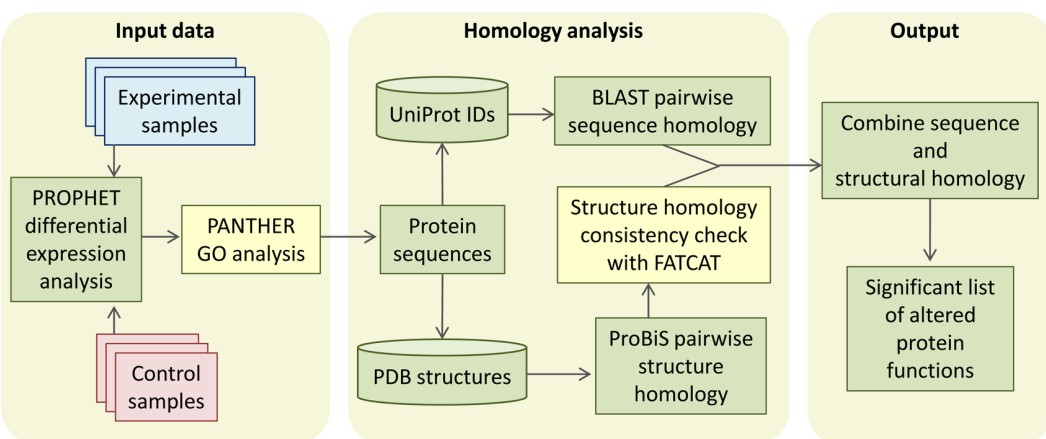

**Figure 1** **The CHURNER workflow.** After differentially expressed proteins are selected and grouped by functional analysis, individual protein sequences are used to retrieve UniProt and PDB identificators. All sequences and structures are compared pairwise, and significant structural alignments are then use to reinforce protein functions that are significantly altered in the experiment. Yellow boxes indicate manual steps.

responses to infection, both in grapevines (*Katam et al., 2015*; *Lin et al., 2007*; *Yang et al., 2011*) and citrus (*Rodrigues et al., 2013*). These include PR proteins, β-1, 3-glucanases, LRR-RLKs and chitinases listed in Table 1. From all proteins detected in xylem sap, those with significant fold changes as indicated by the Prophet/Scaffold program (See File S1 with protein quantification data and File S4 for multifasta protein sequences) were re-annotated using the BLAST command line interface (*Camacho, 2008*). Twelve of those for which a predicted structure could be assigned in the Protein Data Bank were further analyzed in the CHURNER pipeline (Table 2). Despite some UniProt IDs not having the proper functional annotation, as for example protein UID: F6HBN7, which is not annotated as a pathogenesis-related protein in http://www.uniprot.org/uniprot/F6HBN7, CHURNER uses UniProt IDs as protein identifiers, as it is easily streamlined with other applications such as the PANTHER gene ontology analysis server. The PANTHER overrepresentation test uses the most updated gene ontology database released, and has the option to use Bonferroni correction for multiple testing, which did not alter our results. It also has a wide range of reference genomes to be queried with the user list. Nine of twelve of our differentially expressed proteins could be mapped to ontology terms by PANTHER. The molecular functions with significant overrepresentation were chitinase and glycosyl hydrolase activities (GO:0004568, and GO:0016798, respectively). See complete GO analysis in File S5). Accordingly, the biological processes overrepresented were catabolism of glucosamines (GO:1901072), chitin (GO:0006032), amino glycans (GO:0006026) and amino sugars (GO:0046348). Interestingly, identification of differentially expressed proteins demonstrated that NtPRp27, which is found in the xylem exudate of non-infected grapevines (*Agüero et al., 2008*), and is a known pathogenesis-related protein (*Okushima et al., 2000*) was not over-expressed upon *X. fastidiosa* infection. Comparably, two LRR-RLK (Ciclev10004108m and Ciclev10014130m) found to be up-regulated in Ponkan

**Table 1** Proteins identified in grapevine xylem sap of *Xylella fastidiosa*-infected and healthy plants by LC-MS/MS.[a]

| UniProt ID | Protein function | *Arabidopsis* best match | | MW[b] | MP[c] | Cov. (%)[d] | SignalP | TargetP[e] |
|---|---|---|---|---|---|---|---|---|
| | | ID | E-value | | | | | |
| *Pathogenesis-related proteins* | | | | | | | | |
| D7TXF5 | Pathogenesis-related 4 | AT3G04720.1 | 9E−51 | 15 | 6 | 62 | 21\|22 | Sec |
| F6HVL6 | Pathogenesis-related 4 | AT3G04720.1 | 2E−59 | 20 | 2 | 14 | 21\|22 | Sec |
| F6HBN7 | Basic pathogenesis-related 1 | AT2G14580.1 | 1E−47 | 18 | 4 | 37 | 25\|26 | Sec |
| A5BNW5 | Pathogenesis-related 1 | AT2G14610.1 | 2E−53 | 17 | 2 | 23 | 19\|20 | Sec |
| A5AWT7 | Pathogenesis-related thaumatin | AT1G20030.2 | 8E−83 | 45 | 5 | 17 | 26\|27 | Sec |
| A5AWT9 | Osmotin 34 | AT4G11650.1 | 1E−79 | 24 | 13 | 50 | 24\|25 | Sec |
| A5B4P9 | Osmotin 34 | AT4G11650.1 | 2E−82 | 24 | 6 | 47 | 24\|25 | Sec |
| Q9M4G7 | Osmotin 34 | AT4G11650.1 | 4E−72 | 20 | 5 | 42 | – | – |
| Q9M4G6 | Osmotin 34 | AT4G11650.1 | 7E−83 | 24 | 7 | 45 | 24\|25 | Sec |
| A5B2B6 | Osmotin 34 | AT4G11650.1 | 8E−80 | 24 | 3 | 35 | 24\|25 | Sec |
| *Proteases* | | | | | | | | |
| A5AZU5 | Aspartyl protease | AT5G07030.1 | 1E−134 | 40 | 5 | 15 | – | – |
| D7T5Q6 | Aspartyl protease | AT4G35880.1 | 1E−82 | 33 | 2 | 17 | – | – |
| F6H3K5 | Aspartyl protease | AT5G10770.1 | 1E−116 | 46 | 3 | 9 | – | – |
| F6H3K6 | Aspartyl protease | AT5G10770.1 | 1E−152 | 51 | 4 | 11 | 25\|26 | Sec |
| E0CQB3 | Subtilisin | AT1G20160.1 | 0E+00 | 79 | 19 | 36 | – | – |
| F6HNS0 | Subtilisin | AT5G59100.1 | 1E−157 | 69 | 9 | 24 | – | – |
| F6H4J9 | Subtilisin | AT5G67090.1 | 1E−157 | 79 | 2 | 3 | – | – |
| F6HSV1 | Subtilase | AT5G67360.1 | 0E+00 | 81 | 18 | 46 | 24\|25 | Sec |
| F6I357 | Subtilase | AT1G01900.1 | 0E+00 | 81 | 18 | 38 | 20\|21 | Sec |
| A5B179 | Subtilase | AT1G01900.1 | 1E−163 | 75 | 6 | 14 | 20\|21 | Sec |
| A5AIJ0 | Serine carboxypeptidase 20 | AT4G12910.1 | 0E+00 | 54 | 6 | 15 | 26\|27 | Sec |
| A5C816 | Serine carboxypeptidase 51 | AT2G27920.1 | 1E−163 | 51 | 4 | 16 | 21\|22 | Sec |
| *Carbohydrate-active enzymes* | | | | | | | | |
| F6GZC4 | Basic chitinase | AT3G12500.1 | 3E−89 | 35 | 15 | 68 | 21\|22 | Sec |
| Q9ZTK4 | Basic chitinase | AT3G12500.1 | 2E−85 | 35 | 11 | 52 | 21\|22 | Sec |
| A5BK69 | Chitinase A | AT5G24090.1 | 1E−109 | 32 | 12 | 48 | 25\|26 | Sec |
| F6H6H7 | Chitinase A | AT5G24090.1 | 1E−108 | 32 | 9 | 37 | 25\|26 | Sec |
| F6HB09 | Carrot EP3-3 chitinase | AT3G54420.1 | 1E−106 | 29 | 3 | 18 | 29\|30 | Sec |
| O24530 | Carrot EP3-3 chitinase | AT3G54420.1 | 5E−90 | 27 | 2 | 16 | 20\|21 | Sec |
| F6HQS7 | Alpha-L-arabinofuranosidase | AT3G10740.1 | 0E+00 | 84 | 20 | 26 | 28\|29 | Sec |
| F6HLL8 | Beta-1, 3-glucanase 3 | AT3G57240.1 | 1E−109 | 38 | 23 | 69 | 33\|34 | Sec |
| F6HLL9 | Beta-1, 3-glucanase 3 | AT3G57240.1 | 1E−110 | 37 | 4 | 22 | 32\|33 | Sec |
| A7PQW3 | Beta-1, 3-glucanase 3 | AT3G57240.1 | 3E−89 | 37 | 2 | 8 | 29\|30 | Sec |
| F6I6R4 | Beta-D-xylosidase 4 | AT5G64570.1 | 0E+00 | 83 | 23 | 42 | 33\|34 | Sec |
| D7TXW6 | Alpha-galactosidase 2 | AT5G08370.1 | 1E−169 | 45 | 11 | 31 | 24\|25 | Sec |
| F6HGW2 | Beta galactosidase 1 | AT3G13750.1 | 0E+00 | 92 | 2 | 2 | 24\|25 | Sec |
| D7SKW9 | Beta-galactosidase 8 | AT2G28470.2 | 0E+00 | 92 | 2 | 4 | 23\|24 | Sec |
| D7TPI6 | Alpha-amylase-like | AT4G25000.1 | 1E−161 | 47 | 6 | 20 | 22\|23 | Sec |

Table 1 (*continued*)

| UniProt ID | Protein function | *Arabidopsis* best match | | MW[b] | MP[c] | Cov. (%)[d] | SignalP | TargetP[e] |
|---|---|---|---|---|---|---|---|---|
| | | ID | *E*-value | | | | | |
| A5C7G0 | Glucuronidase 3 | AT5G34940.2 | 0E+00 | 71 | 3 | 8 | 18\|19 | Sec |
| F6GU88 | Glycosyl hydrolase | AT5G12950.1 | 0E+00 | 97 | 5 | 8 | 24\|25 | Sec |
| A5AZM8 | Glycosyl hydrolase | AT2G27500.1 | 1E−132 | 50 | 2 | 5 | – | – |
| F6H158 | Glycosyl hydrolase | AT1G58370.1 | 0E+00 | 105 | 5 | 7 | – | – |
| D7SVH6 | Glycosyl hydrolase | AT3G26720.1 | 0E+00 | 114 | 2 | 3 | 19\|20 | Sec |
| D7TQ09 | O-Glycosyl hydrolases | AT4G34480.1 | 1E−171 | 52 | 5 | 11 | 24\|25 | Sec |
| E0CQB9 | O-Glycosyl hydrolases | AT4G34480.1 | 1E−178 | 50 | 3 | 12 | 22\|23 | Sec |
| D7T828 | O-Glycosyl hydrolases | AT5G55180.1 | 0E+00 | 50 | 4 | 17 | 20\|21 | Sec |
| F6HCL5 | Glycosyl hydrolases | AT4G19810.1 | 1E−107 | 40 | 5 | 24 | 25\|26 | Sec |
| D7T548 | Glycosyl hydrolases | AT4G19810.1 | 1E−118 | 40 | 5 | 21 | 25\|26 | Sec |
| A7PZL3 | Pectin lyase-like | AT3G61490.3 | 0E+00 | 53 | 2 | 10 | – | – |
| F6HUM8 | Pectin lyase-like | AT3G61490.3 | 0E+00 | 52 | 7 | 24 | – | – |
| A5AZD0 | Callose-binding protein 3 | AT1G18650.1 | 3E−35 | 20 | 3 | 20 | 19\|20 | Sec |
| D7SI17 | Callose-binding protein 3 | AT1G18650.1 | 3E−39 | 21 | 2 | 20 | 19\|20 | Sec |
| A5C594 | Expansin-like | AT4G17030.1 | 2E−40 | 23 | 6 | 42 | 24\|25 | Sec |
| *Receptor-like kinases (RLKs)* | | | | | | | | |
| F6HIL5 | Receptor-like kinase-related | AT3G22060.1 | 8E−69 | 27 | 16 | 64 | 24\|25 | Sec |
| A5AID0 | Receptor-like kinase-related | AT5G48540.1 | 2E−74 | 45 | 3 | 11 | 25\|26 | Sec |
| D7TPF3 | Leucine-rich repeat (LRR) SHV3-like 2 | AT4G06744.1 | 1E−108 | 49 | 2 | 4 | 28\|29 | – |
| *Peroxidases* | | | | | | | | |
| F6GUF3 | Peroxidase 2 | AT5G06720.1 | 1E−109 | 36 | 2 | 7 | 23\|24 | Sec |
| F6GUE9 | Peroxidase | AT5G19890.1 | 1E−104 | 29 | 15 | 65 | – | – |
| F6HD61 | Peroxidase | AT1G49570.1 | 1E−110 | 36 | 11 | 35 | 25\|26 | Sec |
| A5BJV9 | Peroxidase | AT5G58390.1 | 3E−77 | 28 | 9 | 49 | – | – |
| F6H776 | Peroxidase | AT1G05260.1 | 8E−77 | 74 | 11 | 23 | 21\|22 | Sec |
| F6HIK4 | Peroxidase | AT1G05260.1 | 1E−135 | 76 | 2 | 3 | 26\|27 | Sec |
| D7TQI6 | Peroxidase | AT2G37130.1 | 1E−137 | 37 | 7 | 29 | – | – |
| F6H3X3 | Peroxidase | AT5G14130.1 | 8E−98 | 34 | 5 | 22 | 34\|35 | – |
| D7SVP1 | Peroxidase | AT5G14130.1 | 2E−20 | 10 | 2 | 43 | – | – |
| F6GXY7 | Peroxidase | AT5G05340.1 | 7E−96 | 28 | 2 | 13 | – | – |
| D7SR21 | Peroxidase | AT5G05340.1 | 3E−98 | 28 | 3 | 19 | – | – |
| F6H0Z1 | Peroxidase | AT5G05340.1 | 1E−113 | 34 | 2 | 13 | 22\|23 | Sec |
| A5B8V0 | Peroxidase | AT2G41480.1 | 6E−91 | 30 | 2 | 12 | – | – |
| F6HH88 | Peroxidase | AT2G41480.1 | 1E−120 | 70 | 2 | 7 | 24\|25 | Sec |
| F6HSU5 | Peroxidase | AT5G67400.1 | 1E−138 | 36 | 3 | 17 | 27\|28 | Sec |
| *Others* | | | | | | | | |
| E0CQL6 | Basic blue protein-like | AT2G02850.1 | 8E−38 | 19 | 8 | 49 | – | – |
| D7TML8 | Inhibitor/LTP/seed storage | AT3G53980.2 | 2E−33 | 12 | 11 | 67 | 27\|28 | Sec |
| D7SLG6 | Inhibitor/LTP/seed storage | AT2G44290.1 | 1E−35 | 19 | 3 | 26 | – | – |
| F6H7X9 | Inhibitor/LTP/seed storage | AT4G33550.2 | 1E−07 | 12 | 3 | 39 | 29\|30 | Sec |
| A5C9S3 | Inhibitor/LTP/seed storage | AT4G33550.2 | 4E−05 | 12 | 2 | 25 | 21\|22 | Sec |

**Table 1** (*continued*)

| UniProt ID | Protein function | *Arabidopsis* best match | | MW[b] | MP[c] | Cov. (%)[d] | SignalP | TargetP[e] |
|---|---|---|---|---|---|---|---|---|
| | | ID | *E*-value | | | | | |
| F6I0G4 | Pectin methylesterase inhibitor | AT5G09760.1 | 0E+00 | 61 | 16 | 33 | 21\|22 | Sec |
| A5BS35 | Basic seretory protein | AT2G15220.1 | 1E−83 | 25 | 10 | 53 | 23\|24 | Sec |
| F6HS61 | Glycine-rich protein | AT4G30460.1 | 7E−03 | 13 | 7 | 67 | 22\|23 | Sec |
| A5AIZ1 | Glycine-rich protein | AT4G30460.1 | 3E−03 | 13 | 4 | 67 | 22\|23 | Sec |
| D7TY88 | Protease inhibitor | AT1G17860.1 | 6E−58 | 23 | 5 | 23 | 27\|28 | Sec |
| D7T293 | Cupredoxin | AT4G12420.2 | 0E+00 | 66 | 2 | 5 | 23\|24 | Sec |
| A5BMY7 | Cupredoxin | AT1G72230.1 | 7E−23 | 19 | 2 | 18 | 22\|23 | Sec |
| D7UBD5 | Cupredoxin | AT3G27200.1 | 4E−40 | 18 | 4 | 41 | 23\|24 | Sec |
| A5BZS1 | FAD-binding Berberine | AT4G20840.1 | 1E−179 | 59 | 5 | 10 | 30\|31 | Sec |
| A5B2E1 | Cystatin/Monellin | AT5G47550.1 | 2E−29 | 13 | 2 | 15 | 24\|25 | Sec |
| A5BH21 | PLC-like phosphodiesterase | AT1G66970.1 | 0E+00 | 70 | 3 | 5 | 21\|22 | Sec |
| D7SVW5 | PI-PLC-like | AT4G36945.1 | 1E−143 | 45 | 2 | 9 | 27\|28 | Sec |
| A5BB66 | Fasciclin-like | AT3G60900.1 | 1E−117 | 43 | 3 | 15 | 20\|21 | – |
| A5B7N6 | Fasciclin-like | AT4G12730.1 | 1E−115 | 44 | 2 | 8 | 26\|27 | Sec |
| D7SXH0 | Lamin-like | AT5G15350.1 | 3E−37 | 18 | 2 | 15 | – | – |
| A5AIY9 | Unknown protein | – | – | 15 | 4 | 59 | 23\|24 | Sec |

**Notes.**
[a] A total of 91 proteins with at least two peptides sequenced per protein were identified and are displayed grouped by functional category.
[b] Predicted molecular weight of proteins, in kDa.
[c] Matched peptides.
[d] Percentage of coverage.
[e] TargetP output: Sec, secreted protein; –, undefined.

mandarin infected with *X. fastidiosa* (*Rodrigues et al., 2013*), were not affected in our data, despite LRR-RLKs being detected in our proteomic analysis (Table 1). Plant receptor-like kinases (RLKs) are a large gene family (∼600 members in *Arabidopsis*) (*Shiu & Bleecker, 2001*), consisting of an accelerated evolutionary domain implicated in signal reception through leucine-rich repeat (LRRs) (*Afzal, Wood & Lightfoot, 2008*). Resistance (R) genes have evolved to counter pathogens that bypass the pathogen-associated molecular patterns (PAMP) mechanism in plants (*Nicaise, Roux & Zipfel, 2009*). Most R genes encode proteins comprising of a nucleotide-binding site (NBS) and leucine-rich repeats (LRRs), and recognize and neutralize specialized pathogen avirulence (Avr) proteins, providing plants with resistance (*Borhan et al., 2004*; *Chakraborty et al., 2016*; *Ernst et al., 2002*; *Hayashi et al., 2010*; *Zhang et al., 2010*).

## Up-regulated proteins

The up-regulated protein F6HLL8 listed in Table 2 is a $\beta$-1, 3-glucanase (GNS), a well-established pathogenesis related protein (*Balasubramanian et al., 2012*; *Shinshi et al., 1988*). GNS has strong anti-microbial (*Xie et al., 2015*) and anti-fungal activity (*Su et al., 2013*). Expectedly, it is a target of pathogen toxins in the ensuing evolutionary battle (*Sanchez-Rangel, Sanchez-Nieto & Plasencia, 2012*; *Zhang et al., 2015b*). GNS, along with chitinase, has been shown to inhibit fungal growth (*Mauch, Mauch-Mani & Boller, 1988*; *Sela-Buurlage et al., 1993*). The presence of two up-regulated chitinases (UIDs: A5BK69,

**Table 2  Differentially expressed proteins and their sequence and structural similarities with reference proteins.**

| UniProt ID | Protein function | GenBank / protein data base | | | Expression[a] |
|---|---|---|---|---|---|
| | | ID | Score | E-value | |
| F6HLL8 | $\beta$-1-3 glucanase | NP_001268153.1 / 4HPG | 696 / 423 | 0e+00 / 1e−147 | Up |
| A5BK69 | Class III chitinase | ACH54087.1 / 1HVQ | 592 / 435 | 0e+00 / 1e−153 | Up |
| D7T548 | Chitotriosidase-1 | XP_002270368.1 / 3AQU | 743 / 452 | 0e+00 / 5e−158 | Up |
| F6HBN7 | Pathogenesis-related | XP_002274275.1 / 1CFE | 334 / 184 | 1e−114 / 3e−59 | Up |
| A5BNW5 | Pathogenesis-related | XP_002273788.2 / 1CFE | 333 / 186 | 7e−114 / 4e−60 | Up |
| D7TML8 | Lipid-transfer protein | XP_002281554.1 / 2RKN | 239 / 36.2 | 9e−79 / 7e−04 | Up |
| A5C594 | Expansin-like | XP_002270175.2 / 2HCZ | 374 / 89 | 2e−128 / 5e−21 | Up |
| E0CQL6 | Blue copper protein-like | XP_002266573.1 / 2CBP | 268 / 158 | 4e−89 / 3e−49 | Up |
| F6GZC4 | Chitinase | CAC14015.1 / 4DWX | 660 / 385 | 0e+00 / 7e−134 | Down |
| A5AWT7 | Thaumatin-like 1b | XP_002274137.1 / 3ZS3 | 518 / 253 | 1e−180 / 3e−81 | Down |
| F6I6R4 | $\beta$-xylosidase | XP_002264183.2 / 1EX1 | 1605 / 154 | 0e+00 / 2e−39 | Down |
| D7SLG6 | YLS3-like, Lipid-transfer protein | XP_002285691.1 / 1FK0 | 355 / 33.9 | 3e−122 / 1.7e−02 | Down |

Notes.
[a]Variation of protein level detected in sap from infected grapevine compared to uninfected control.

D7T548) induced by *X. fastidiosa* demonstrates that this defense response is similar during both bacterial and fungal attack. Both these chitinases belong to the GH18 sub-family of chitinases (*Funkhouser & Aronson Jr, 2007*). Another up-regulated protein is an expansin-like B1 enzyme (UID: A5C594), also a carbohydrate-binding protein involved in cell-wall loosening and restructuring (*Zhang et al., 2014*). Homologs have previously been link to abiotic stress responses (*Han et al., 2014*; *Nanjo, Nakamura & Komatsu, 2013*). Two other proteins (UID: F6HBN7, A5BNW5) are the well-established defense PR-1 pathogenesis related proteins (*Sudisha et al., 2012*). Several functions have previously been attributed to PR1 including, antiviral activity in tobacco (*Antoniw & White, 1980*) and anti-herbivory activity in maize (*Zhang et al., 2015a*). Additional activities suggest protease-mediated programmed cell death pathways in plants (*Lu et al., 2013*), a symptom commonly seen in leaves of Pierce's diseased grapevines. A lipid-transfer protein of the DIR1 type (UID: D7TML8) also ranks among the up-regulated proteins, suggesting a systemic defense response being activated, as previously seen in *Arabidopsis* (*Champigny et al., 2013*). A blue copper protein-like was also identified as up-regulated. This redox protein family is part of a widespread but yet poorly characterized defense mechanism in plants and/or lignin formation (*Cao et al., 2015*; *Nersissian et al., 1998*). It is interesting to note an abundance of cell-wall modification enzymes in xylem sap from infected plants, as this corroborates previous observations in other pathosystems such as *Xanthomonas oryzae* infection of rice (*Hilaire et al., 2001*), among others (*Van Loon, Rep & Pieterse, 2006*). Since vines infected with *X. fastidiosa* commonly display an increase in stem diameter, we verified if this reflected in an increase of secondary wall deposition by nuclear magnetic resonance imaging (Fig. 2). Indeed the MRI showed an increase in dense material (colored in light shades in Fig. 2) in infected plant stems, including those used for xylem sap collection. Recently it has also

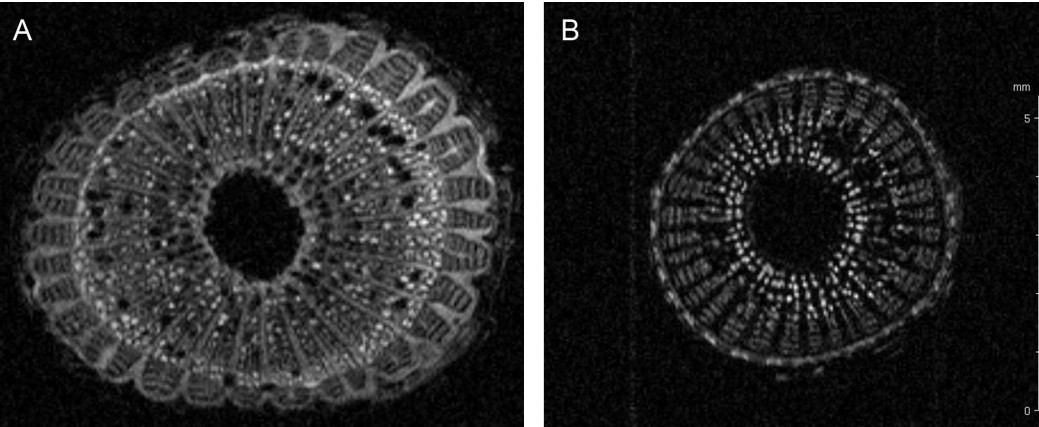

**Figure 2** **Cell wall thickening of infected grapevines.** Magnetic resonance imaging (1H-MRI) of stems of *Xylella fastidiosa* (A) infected and (B) non-infected grapevines. Note the brighter contrast (denser material) of secondary xylem and phloem vessels on the infected vine. Both images have the same magnification and scale bar is 5 mm. Images representative of transversal stem cuts obtained near 10 cm from top central stem. Similar thickening of cell walls were also observed in transversal cuts obtained along the vine until the base near the soil.

been demonstrated that citrus infected with *X. fastidiosa* display a thickening of secondary cell-walls (*Niza et al., 2015*).

## Down-regulated proteins

As mentioned previously the GH19 sub-family chitinase (UID: F6GZC4) is down-regulated. A similar suppression of a chitinase gene in response to mycorrhizal fungus *Glomus intraradices* infection of tobacco roots has been noted previously (*David et al., 1998*). The GH19 sub-family has members that are sugar-binding proteins but without catalytic activity (*Martinez-Caballero et al., 2014*). Further experimentation is necessary to verify whether this is the case in *Vitis vinifera*, as it can be part of the cell-wall remodeling in response to the pathogen, as previously suggested by other works (*Lin et al., 2007*; *Rodrigues et al., 2013*). A thaumatin-like protein (TLP) (UID: A5AWT7) is also found to be down-regulated upon *X. fastidiosa* infection. TLP's are found in most eukaryotes, and involved in host defense and several developmental processes (*Liu, Sturrock & Ekramoddoullah, 2010*). While TLP over expression has been shown to enhance resistance to *Alternaria alternata* in tobacco (*Safavi, Zareie & Tabatabaei, 2012*), these proteins can also be down-regulated in some cases, as shown in the compilation of transcriptomes of poplar leaf rust infections (*Petre et al., 2011*). A β-D-xilosidase 4 ortholog (UID: F6I6R4) was also down-regulated in infected vines, again reinforcing the drastic effect on cell-wall remodeling enzymes upon pathogen infection. The other down-regulated protein in Table 2 is a LTP similar to YLS3 (yellow-leaf-specific) which is a marker of leaf senescence, being induced in earlier stages and repressed in later stages in *Arabidopsis* (*Yoshida et al., 2001*).

## Sequence homology

Although in our data set only twelve proteins were selected for further analysis through CHURNER, there might be cases when larger data sets render a manual inspection of

**Table 3  Pairwise BLAST[a] results of example proteins analyzed with CHURNER.**

| UID[b] #1 | UID #2 | Identity[c] | E-value | Protein function |
|-----------|--------|-------------|---------|------------------|
| F6HBN7 | A5BNW5 | 120/136 | 8e−90 | Pathogenesis-related proteins (PR-1) |
| D7TML8 | D7SLG6 | 20/78 | 9e−08 | Lipid transfer proteins |
| A5BK69 | A5C594 | 30/181 | 1.6e−2 | Chitinase or expansin (?) |
| F6HBN7 | A5BK69 | 16/133 | 2.4 | PR-1 or Chitinase (?) |
| A5BK69 | D7T548 | 36/272 | 2e−3 | Chitinases GH18 |

Notes.
[a] Sequences aligned with bl2seq from NCBI.
[b] UniProt identificator.
[c] Total number of identical amino acid residues considering the best alignment between the two proteins.

homologous proteins difficult. Even in the current case, it is difficult to identify D7SLG6 as a lipid transfer protein (LTP) from the BLAST automated annotation (Table 2, value marked in italic). Thus, as the next step in CHURNER, we implemented a pairwise BLAST of all mature proteins (devoid of signal sequences). Table 3 shows the pairwise sequence homology with an E-value cutoff of 0.005. There are several interesting aspects that emerge from this comparison. As expected, the two PR-1 proteins are found to be significantly homologous. The 'YLS3-like' protein (UID: D7SLG6) is found to be quite similar to another LTP (UID: D7TML8). Furthermore, we observe sequence homology between chitinases and expansin (E-value = 4e−04), and much less between chitinases and PR-1 proteins (E-value = 0.002). Interestingly, these similarities are greater than that between the two known chitinases from sub-family GH18 (E-value = 0.003). This raises the interesting question whether these proteins (chitinases/expansin/PR-1) have promiscuous functions (*Chakraborty & Rao, 2012*; *Khersonsky & Tawfik, 2010*), and underlines the problem of depending only on annotation of individual protein sequences, thus providing a more rational alternative to identify proteins with similar functions. Since the structure of a protein is intrinsically related to its function, we implemented structural annotation as the next step in CHURNER.

## Structural annotation

CHURNER implements an automated search for homologous proteins with known PDB structures (Table 2). As expected, proteins with high similarity and alignment at the sequence level map to the same PDB structure, such as the PR-1 proteins (PDBid: 1CFE, PR-14a protein). Interestingly, this protein was identified as a possible replacement of the human neutrophil elastase component of the chimeric protein (*Chakraborty, 2012*; *Chakraborty et al., 2013*) that provided enhanced grapevine resistance to *X. fastidiosa* (*Dandekar et al., 2012*). Moreover, apart from the LTP (UID: D7SLG6, E-value = 0.017), all matches are very significant to their PDB closest model.

## Structural homology

Using the structures corresponding to the identified proteins, we detected similarities that might have escaped detection in the sequence homology search. Table 4 shows the most significant pairwise structural comparison of the structures (excluding the PR-1

**Table 4** Pairwise superimposition of the PDB structures using ProBiS.

| PDBid/UID[a] #1 | PDBid/UID #2 | PZ[b] | Protein function | FCTSIG |
|---|---|---|---|---|
| 2RKN/D7TML8 | 1FK0/D7SLG6 | 2.4 | Lipid transfer proteins | Yes |
| 1HVQ/A5BK69 | 3AQU/D7T548 | 2 | Chitinases GH18 | Yes |
| 3AQU/D7T548 | 1EX1/F6I6R4 | 1.6 | Chitinase and $\beta$-D-glucan exohydrolase | Yes |
| 1HVQ/A5BK69 | 1EX1/F6I6R4 | 1.3 | Chitinase and $\beta$-D-glucan exohydrolase | No |

**Notes.**
[a] UID: UniProt identificator.
[b] The results are sorted based on the ProBiS ZScore (PZ). Significance of structural alignment was verified using FATCAT (FCTSIG). Although the two chitinases have no sequence homology, their structural features are conserved. Similarly, we see signifiant structural similarity between a chitinase (UID: D7T548) and a $\beta$-D-Glucan Exohydrolase (UID: F6I6R4).

proteins which have the same PDB structure) computed using ProBiS (*Konc & Janezic, 2010*). We subsequently verified the alignment significance using the FATCAT server (*Ye & Godzik, 2004*). The ProBiS Z-score (PZ in Table 4) are standardized alignment scores (*Konc & Janezic, 2010*) which provide statistical and structural significance of local structural alignments. Z-scores >2 are considered highly significant (PDBs: 2RKN/1FK0 and 1HVQ/3AQU), although a Z-score of 1.6 is also significant (PDBs: 3AQU/1EX1), as confirmed by FATCAT (which looks at the global structure). The LTPs (with a sequence homology $E$-value $= 1e-06$) are structurally homologous, as expected (Fig. 3A). Noteworthy, the chitinases from the GH18 family with a low sequence homology ($E$-value $= 0.003$) are structurally homologous (Fig. 3B). Finally, in spite of a much lower sequence homology ($E$-value $=0.44$), the chitinase (UID: D7T548) and the $\beta$-D-glucan exohydrolase (UID: F6I6R4) are structurally homologous (Fig. 3C). These observations highlight the necessity of structural comparison in annotating and grouping proteins based on functionality in proteomic analysis, and points to alternative protein functions that can be tested in subsequent studies.

Environmental stimuli (*Yang et al., 2012*), pathogens (*Moy et al., 2004*) or disease (*Ng et al., 2009*) induce differential expression of specific genes. Rapid technological advances have helped us identify these genes, and define their roles in defense or pathogenesis. While quantifying transcripts through high-throughput sequencing techniques have revolutionized these efforts (*Wang, Gerstein & Snyder, 2009*), the correlation between transcriptional and protein abundance remains suspect (*Gygi et al., 1999*) due to the complexity of the regulatory factors modulating translation (*Zhu et al., 2012*). Thus, identifying proteins through techniques like mass spectrometry (*Witzel et al., 2009*), and measuring their relative amounts (*Hu, Rampitsch & Bykova, 2015*), provides a true picture of the genes involved in pathogenesis and defense response, rather than measuring their RNA abundance (*Moy et al., 2004*). Nevertheless, the value of transcriptomic analysis should not be underestimated as it has proved to be an effective approach to discover genes responsive to infection, as exemplified by sequencing of expressed sequence tags from *X. fastidiosa*-infected grapevines (*Lin et al., 2007*) and citrus (*Rodrigues et al., 2013*).

Here we showed that several proteins (pathogenesis-related PR-1, chitinases and $\beta$-1, 3-glucanases) are differentially expressed in the xylem sap of grapevine infected with *X. fastidiosa*, using LC-MS/MS with at least two peptides sequenced per protein, confirming

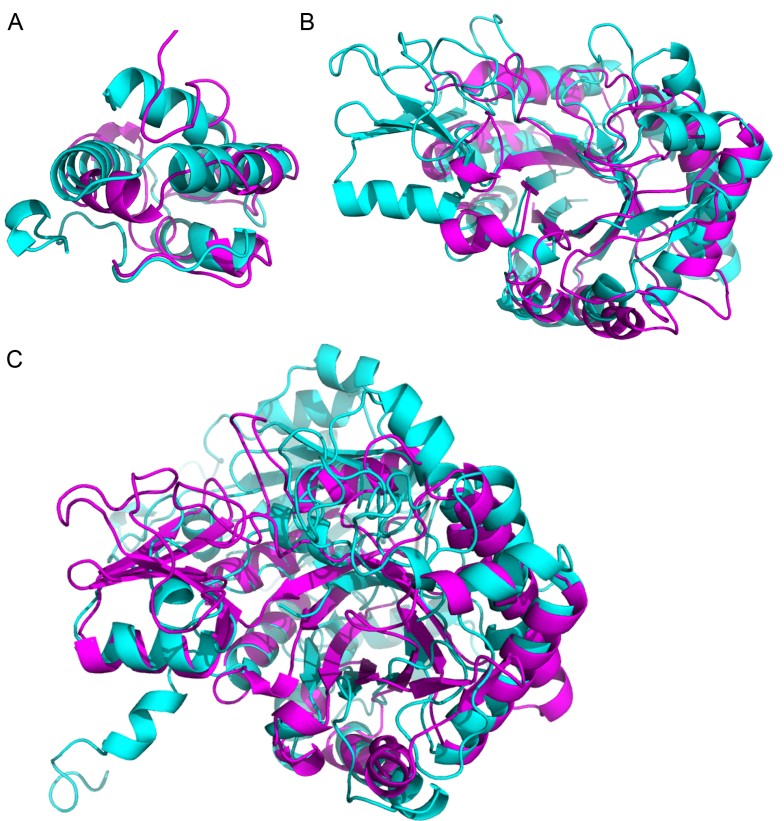

**Figure 3** **Superimposition of proteins that have significant structural homology.** Structural homology has been detected using ProBiS, and confirmed using FATCAT. (A) Lipid transfer proteins: PDBid: 2RKNA (in magenta) and PDBid: 1FK0A (in cyan). (B) Chitinase GH18 proteins: PDBid: 1HVQA (in magenta) and PDBid: 3AQUA (in cyan). Note, that these proteins have low sequence homology (BLAST $E$-value = 0.003). (C) Chitinase (PDBid: 3AQUA, in magenta) and $\beta$-D-Glucan Exohydrolase (PDBid: 1EX1A, in cyan). These proteins have low sequence homology ($E$-value = 0.44).

findings of previous investigations. These proteins are recognized as key players in the plant response against pathogen infection, as exemplified by sugarcane infected by *Sporisorium scitamineum* (*Su et al., 2013*) and other pathosystems reviewed in (*Sudisha et al., 2012*). Interestingly, we observed that there are two chitinases which are regulated differently (UniProt IDs F6GZC4 and D7T548, Table 4). Chitinases degrade chitin, a component of fungal cell walls (*David et al., 1998*). This observation diminishes the importance of chitinase as a generic defense agent against *X. fastidiosa*. It is also possible that the plant response to the infecting bacterial agent is indifferent to the expression levels of these anti-fungal chitinases, for which we warrant further studies. The pathogenic state can also be characterized by differentially expressed genes within *X. fastidiosa* itself (*Shi et al., 2010*). Consequently, the analysis of differentially expressed genes and proteins should consider functionally related proteins as a single entity, and not just the expression levels of single genes or proteins. CHURNER allows for such selection based on both sequence and structural homology, as often, the best match for a sequence has an incomplete annotation. Subsequently, structural homology can identify functional relationship among

**Table 5  Potential and spatial congruence of the active site residues in proteins chitinase and β-D-glucan exohydrolase detected using CLASP.**

| PDB[*] | Active site atoms (a, b, c) | | ab | ac | bc |
|---|---|---|---|---|---|
| 1EX1A | ASP285 OD1, GLU491 OE1, TRP286 CZ2 | D | 6.4 | 8.4 | 9.3 |
| | | PD | −24.1 | −273.2 | −249.1 |
| 3AQU | GLU116 OE1, ASP114 OD1, TRP324 CZ2 | D | 7.1 | 7.9 | 9.5 |
| | | PD | 49.8 | −296.8 | −346.7 |

**Notes.**
[*]Chitinase: 1EX1A, β-D-glucan exohydrolase: 3AQU. The ability of CLASP to select stereo-chemically equivalent residues (Asp and Glu, both negatively charged residues) is critical to find the homologous active site. D, Pairwise distance in Å. PD, Pairwise potential difference. See Methods section for units of potential.

sequences with little sequence homology. For example, the two chitinase homologs have low pairwise sequence homology ($E$-value = 0.003). However, their significant homologous counterparts in the PDB database are PDBid: 4DWXA (*Secale cereale*, rye) and PDBid: 3AQU (*Arabidopsis thaliana*) have significant structural homology, as computed using ProBiS (*Konc & Janezic, 2010*) and FATCAT (*Ye & Godzik, 2004*).

## Corroborating the promiscuity of the chitinase and β-D-glucan exohydrolase by active site structural homology

The β-D-glucan exohydrolase (PDBid: 1EX1) is critical for hydrolysis of cell walls, containing high levels of 1, 3-β-D-glucans, during wall degradation in germinated grain and during wall loosening in elongating coleoptiles (*Varghese, Hrmova & Fincher, 1999*). Interestingly, the fungal wall is composed of chitin, 1, 3-β- and 1, 6-β-glucan, mannan and proteins (*Adams, 2004*). Thus, the up-regulation of both chitinases and β-D-glucan exohydrolases is possibly an anti-fungal defense response that has been triggered by *X. fastidiosa*. We have seen that the chitinase and the β-D-glucan exohydrolase have no sequence homology (Table 3), but partial structural homology (Table 4). A detailed analysis of their catalytic residues further strengthens credence of their functional similarity. For β-D-glucan exohydrolase, Asp285 and Glu491 are involved in catalysis (*Varghese, Hrmova & Fincher, 1999*). In chitinases, the Asp114 and Glu116 are a part of the conserved motif (DXXDXDXE) (*Hamid et al., 2013*). The active site residues of these proteins demonstrate significant spatial (Fig. 4) and electrostatic congruence (Table 5) determined using CLASP. The absence of sequence linearity indicates that this homology arose from convergent evolution. Remarkably, based on the catalytic triad of the barley β-D-glucan exohydrolase (PDBid: 1EX1A) provided to CLASP, it was able to pick up the catalytic triad from the *Arabidopsis* chitinase (PDBid: 3AQU) despite the lack of sequence homology. We later realized this prediction was correct consulting the work from Ohnuma and collaborators (*2011*) on the crystallographic studies of this type V chitinase.

## CONCLUSIONS

A robust proteomic methodology was used to identify novel protein fragments in addition to the proteins identified previously using two-dimensional gel electrophoresis followed by sequencing of protein spots in non-infected xylem exudates collected from *Vitis*

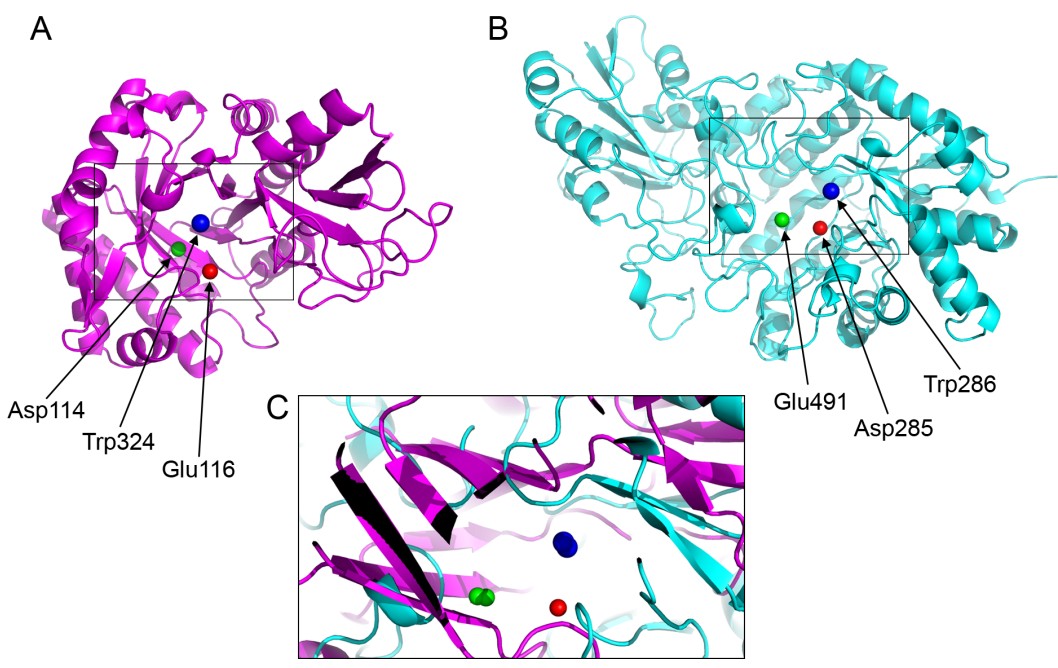

**Figure 4** **Active site residues in the chitinase and β-D-glucan exohydrolase.** (A) Glu116 (red), Asp114 (green), Trp324 (blue) in chitinase (PDBid:3AQU). (B) Asp285 (red), Glu491 (green), Trp286 (blue) in β-D-glucan exohydrolase (PDBid:1EX1A). (C) Superimposition of the chitinase (in magenta) and β-D-glucan exohydrolase (in cyan).

*vinifera* cv. Chardonnay vines (*Agüero et al., 2008*), and mass spectrometric analysis of *X. fastidiosa*-infected grapevines (*Katam et al., 2015*; *Yang et al., 2011*). Thus, here we link the expression pattern of well-studied proteins in grapevines to the pathogen perception response, and present a methodology for assessing their significance by taking into account both their sequence and structural information. Although our data came from a single round of infection containing three plants (and three non-infected control plants), our findings are consistent among the different samples and to previous proteomic studies of *X. fastidiosa* infected grapevines. In conclusion, CHURNER enhances our ability to find functionally-relevant protein candidates that have little or no sequence similarity, and thus would be considered as separate components of a data set. The name "CHURNER" was inspired in the mixing tool used to reach the "cream." We intend to offer a tool to enable for detection of "cream" protein functions, not obvious from simple amino acid sequence alignments. The reduced data set employed in this work was used as a proof of concept, and we encourage readers to use complex data sets with thousands of proteins to find many putative functional relations among proteins that are yet unexplored.

## ACKNOWLEDGEMENTS

We thank James Mitch Elmore for helpful suggestions in proteomic analysis.

### Funding

AMD received grant support from the California Department of Food and Agriculture PD/GWSS Board. BJR received financial support from Tata Institute of Fundamental Research (Department of Atomic Energy). Additionally, BJR received the JC Bose Award Grant from the Department of Science and Technology. RN, LRG, and PAZ received funding from the Brazilian Ministry of Science (CNPq) and Education (CAPES). The funders had no role in study design, data collection and analysis, decision to publish, or preparation of the manuscript.

### Grant Disclosures

The following grant information was disclosed by the authors:
California Department of Food and Agriculture PD/GWSS Board.
Tata Institute of Fundamental Research (Department of Atomic Energy).
JC Bose Award Grant from the Department of Science and Technology.
Brazilian Ministry of Science (CNPq) and Education (CAPES).

### Competing Interests

The authors declare there are no competing interests.

### Author Contributions

- Sandeep Chakraborty conceived and designed the experiments, performed the experiments, analyzed the data, wrote the paper, prepared figures and/or tables, reviewed drafts of the paper.
- Rafael Nascimento and Hossein Gouran performed the experiments, analyzed the data, wrote the paper, prepared figures and/or tables, reviewed drafts of the paper.
- Paulo A. Zaini analyzed the data, wrote the paper, prepared figures and/or tables, reviewed drafts of the paper.
- Basuthkar J. Rao and Luiz R. Goulart contributed reagents/materials/analysis tools, reviewed drafts of the paper.
- Abhaya M. Dandekar conceived and designed the experiments, analyzed the data, contributed reagents/materials/analysis tools, wrote the paper, reviewed drafts of the paper.

### Data Availability

   Data can be found at Zenodo: http://dx.doi.org/10.5281/zenodo.50672.

### Supplemental Information

Supplemental information for this article can be found online at http://dx.doi.org/10.7717/peerj.2007#supplemental-information.

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
