# Peer review of "Sequence/structural analysis of xylem proteome emphasizes pathogenesis-related proteins, chitinases and β-1, 3-glucanases as key players in grapevine defense against Xylella fastidiosa"

_PeerJ, doi:10.7717/peerj.2007_

## Round 0.1 · original submission · Major Revisions

There are several improvements suggested by the two reviewers - please address them in your revision and detail your changes in your rebuttal letter.

Reviewer 1 ·

Basic reporting

The article submitted by Chakraborty et al. entitled "Sequence/structural analysis of xylem proteome emphasizes pathogenesis-related proteins, chitinases and β-1,3-glucanases as key players in grapevine defense against Xylella fastidiosa" is adequately written and contains useful information to the plant pathology and physiology research community.

In terms of basic reporting, areas that require attention are:

1. The overall focus of the paper is somewhat divided. In some cases, the tone of the paper appears focused on the development of a proteomics pipeline using Xf-infection as an example dataset (Abstract + Results & Discussion), while in the introduction the material is strictly related to information of Xf-infection with minimal information of the need for an improved proteomics pipeline. The authors should consider ways to improve the overall focus of the manuscript, perhaps by including a paragraph in the introduction to explain why they developed CHURNER.

2. The sentence in "lines 57-63" is rather long and complicated. Separate the information into two sentences.

3. There is minimal comparison of the data set obtained in this study to other xylem sap proteomes obtained from pathogen-infected plants. The authors provide comparable xylem sap proteomics studies (lines 65-66 -- although Floerl et al. 2008 is focused on apoplast fluid rather than sap). Are glucanases commonly observed in xylem sap in those studies? The manuscript would benefit from this comparison. I would also direct the authors to two recent pathogen-xylem sap proteomics studies (Gawehns et al. 2015 doi: 10.3389/fpls.2015.00967 ) (Pu et al. 2016 doi: 10.3389/fpls.2016.00031)

4. The first paragraph in the Results & Discussion section (lines 67-80) appears to be better suited for the final paragraph of the introduction as it identifies an area of need for proteomics research and the goes on to describe the results of the entire study.

5. Line 78 mentions the recalibration of fold changes, however the results described in this study don't appear to be quantitative as no fold-changes are described in the text. The table describing the differentially abundant proteins should include the fold changes present in the supplementary information (Supplementary File 1).

6. Supplementary information should be referenced in the text to direct the reader to the appropriate files (Line 144)

7.The last sentence of the paragraph “Differentially expressed proteins in xylem sap of infected plants…” starting on line 178 isn’t necessary.

8. It is not clear why the authors describe UID:F6HLL8 as the “first protein” in table 2 (line 182). Is there any significance to this?

9. The term “abiotic defense responses” is not consistent with terminology used by the majority of researchers in the field. Consider adjusting this to “abiotic stress responses”

10. On lines 194-196 the authors indicate that the function of the PR1 protein is unknown. There are studies that demonstrate the antifungal activity of tobacco PR1 (Antonew et al. 1980) and the anti-herbivory activity of maize PR1 (Zhang et al. 2015; doi: 10.1186/s12864-015-1363-1)

11. Line 197 should ready “DIR1-type” rather than “DIR-1 type”.

12. There is very little introduction to the data presented in Figure 2. If the authors feel that this data is important to support their proteomics results, then a proper introduction to this figure, which includes the rationale for the experiment, should be included in this manuscript.

13. On lines 208-209 the authors state “A similar suppression of a chitinase gene in response to fungal co-localization has been noted previously (David et al. 1998)”. The authors should provide more detail on this experiment as it pertains to the relevance of the comparison. Did the David et al. study also use Xf-infected xylem sap or leaves? This information as well as the host and pathogen used by David et al should be clearly stated. The authors should similarly clarify the details of the “Safavi et al. 2012” reference used on line 215, as well as the references on line 249 (Chakraborty 2012; 2013).

14. Consider being more clear at the beginning of the “Seqeunce Homology” paragraph – lines 223-226. Perhaps separate these thoughts into two distinct sentences rather than using brackets.

15. Consider including a value of % amino acid similarity in Table 3 to make the data on protein similarly more accessible to a larger readership. It may be easier for readers understand % AA similarity compared to E. values from BLAST results.

16. Provide rationale for using ProBis and FATCAT in the “Structural Homology” section. It would be beneficial to have a more fleshed out understanding of these programs, as you make claims of structural similarity in the text. Perhaps mention the PZ scores (table 4) in text.

17. The sentence “These proteins are recognized as key players in the plant defense response” on line 278 requires clarification. What defense response are the authors referring to? What host(s)? What type of pathogen? etc

18. The conclusion section starts with new information that appears better suited to be it’s own paragraph entirely (perhaps focused on comparing the xylem sap proteome of this study to other published works) as it brings forth new information on the analysis of xylem sap in untreated grapevines.

19. The authors cannot claim to have identified “functionally-relevant” protein candidates (line 308) as they do not provide any evidence that these proteins are required for (or function in) the defense response to Xf infection.

Experimental design

1. The authors must include methods pertaining to Figure 2 in the Materials and Methods section. It would also be useful to add some description of the scale used in the figures. Are both (A) and (B) pictures to the same scale?

2. It would be useful to include the # of peptides identified per protein in Table 1.

3. Additional information on how proteins were quantified (LC-MSMS) would be useful. Either in the methods section of in the text in the Results and Discussion section.

4. A potential flaw in the experimental design for the comparison of protein sequence homology (Table 3) is the inclusion of the divergent protein secretion (ER) signal sequences. The study would benefit from calculating the effect of the signal sequence on each comparison by including E-values (or % amino acid sequence similarity as suggested in the “Basic Reporting” section) of proteins with and without the predicted signal sequence. The degree of protein similarity will likely increase when comparing the mature protein sequences lacking the signal peptide. This is important as the crystal structures provided in Figure 3 (used to make structural comparisons) were likely derived from mature proteins lacking the these peptides. The authors already provide info on where the signal peptide is located, so this should not be too hard to accomplish. If the authors have already accounted for the signal peptide in this analysis, then this needs to be clearly stated in the methods section as well as the results and discussion section where the data is described.

5. The authors claim to have detected Xf in the xylem sap of infected plants in the materials and methods section but do not provide any evidence for this (line 87-88) ” The presence of Xf in the xylem sap of infected plants was confirmed using anti-Xf antibodies in a Double Antibody Sandwich ELISA (Agdia, USA) following manufacturer’s instruction”. The Xf-detection data should be included as a supporting figure and mentioned in the results and discussion.

6. The authors state “Xylem sap (30-50 mL per plant) was collected overnight in the second week of spring by drip of the cut stem of Xf-infected and non-infected plants” in the materials and methods (lines 89-90). Is it important for the reader to know the location of the stem on the plant used for xylem sap collection? i.e between such and such internode, or even “towards the top or bottom of the plant”.

Validity of the findings

1. The amount of protein obtained in Xf-infected vs. non-infected plants should be stated somewhere in the manuscript, alongside the method(s) used for this determination (i.e. BCA assay, Bradford assay). More importantly, the amount of protein used for proteomics analysis must be stated, as it is currently not clear whether the authors analyzed equal amounts of protein from infected vs. control plants, which brings the validity of their quantitative data into question.

2. The proteomics data was obtained from a single experimental replicate of infected vs. non-infected plants growing over the course of 1 year (post Xf-infection). I appreciate the difficulty in conducting such a long-term experiment, however the authors should acknowledge somewhere in the text that the significance of their study is limited by the analysis of 1 experimental replicate. This fact certainly doesn’t prevent the study from being publishable, however it is important for the readers to understand this limitation. If the authors do not agree, then they must explain why a single replicate is sufficient in the Results and Discussion.

3. The number of experimental replicates for the results displayed in Figure 2 is not clear and should be stated.

Reviewer 2 ·

Basic reporting

This manuscript reports studies which have used a variety of complementary tools to examine proteins differentially expressed. The authors named the mix of tools used to analyze the data as “CHURNER”. The authors point out that structural data should be incorporated in the pipeline of proteomic studies to make inferences about protein function, since some proteins can have similar structures but not necessarily have sequence homology. To demonstrate the importance of this analysis the authors used xylem sap from grape infected or not with Xylella fastidiosa (a plant pathogen bacterium that colonize xylem vessels in plant host), to compare the differential expression of proteins and apply CHURNER. In general this study adds interesting information of how analyze and interpret the function of proteins in approaches originated from proteomic data. On the other hand, there is a lack of knowledge about studies on Xylella fastidiosa that have to be included to improve the discussion of the manuscript.

1. Concerning the proteins found and emphasized in the manuscript (PR-1, chitinases and B-1,3-Glucanase) as important to grapevine defense against X. fastidiosa, there is a lack of knowledge about studies of gene expression in X. fastidiosa over the years, that have described and explained already most of the findings presented here-although with different techniques. For instance Lin et al., 2007 (BMC Plant Biology Journal) found many genes involved in grape responses to X. fastidiosa infection, including those encoding PR proteins, B-1,3 glucanases and chitinases. In addition, more recently Rodrigues et al (2013, BMC Genomics), also found chitinase (expansin) and other genes (including PR-1 in time course of infection) involved in citrus responses to X. fastidiosa. Observe that in the line 211 the authors mentioned “ …it can be part of the cell-wall remodeling in response to the pathogen”, the similar hypothesis was already done by Lin et al (2007)….. “Such cell wall modifications have been hypothesized to be physical barriers to limit further pathogen invasion” and reinforced by Rodrigues et al (2013) that concluded “This work demonstrated that the defense response to the perception of bacteria involves cell wall modification…”. These are only some examples of many other statements done in the manuscript already published by other authors, but no mention of these previous studies was included in discussion.

2. Reading the previous studies, it seems that there is a good correlation about the hypothesis of host (grape and citrus) defense response against X. fastidiosa, among the data obtained using gene expression and protein expression showed in this work. However, in general, the authors mentioned the opposite (lines 266-275). I suggest the authors use the data already published for this pathogen to reinforce the data obtained in this manuscript. In this way CHURNER analysis could be shown as a power tool to improve the knowledge of the gene/protein for further investigation in functional approaches such as transgenic, breeding and others.

Experimental design

In MM section: Xylem sap collection and protein precipitation.
More information should be added in this section. Were those plants kept in field condition? The plants showed PD symptoms? If so, in which stage? How did you get 30-50 ml of xylem sap using only three plants? By drip of cut stem? Are you sure of this information? Please explain better this methodology, since high amount of xylem sap is very difficult to be obtained.

Validity of the findings

4Results and Discussion. Lines 156 – 158. In table 1 is shown many genes that are located in the plant cell, such as RLKs (receptor like kinases). How to explain these proteins be found in xylem sap?
Lines 199 – 204. Note that similar observation, where modification of cell wall by lignification of xylem cells in response to X. fastidiosa were also recently showed by Niza et al., 2015 (Plant Pathology). In addition there other related work done by research groups led by Labavitch and Walker that could be also used to improve the discussion.

Additional comments

This manuscript reports studies which have used a variety of complementary tools to examine proteins differentially expressed. The authors named the mix of tools used to analyze the data as “CHURNER”. The authors point out that structural data should be incorporated in the pipeline of proteomic studies to make inferences about protein function, since some proteins can have similar structures but not necessarily have sequence homology. To demonstrate the importance of this analysis the authors used xylem sap from grape infected or not with Xylella fastidiosa (a plant pathogen bacterium that colonize xylem vessels in plant host), to compare the differential expression of proteins and apply CHURNER. In general this study adds interesting information of how analyze and interpret the function of proteins in approaches originated from proteomic data. On the other hand, there is a lack of knowledge about studies on Xylella fastidiosa that have to be included to improve the discussion of the manuscript.

1. Concerning the proteins found and emphasized in the manuscript (PR-1, chitinases and B-1,3-Glucanase) as important to grapevine defense against X. fastidiosa, there is a lack of knowledge about studies of gene expression in X. fastidiosa over the years, that have described and explained already most of the findings presented here-although with different techniques. For instance Lin et al., 2007 (BMC Plant Biology Journal) found many genes involved in grape responses to X. fastidiosa infection, including those encoding PR proteins, B-1,3 glucanases and chitinases. In addition, more recently Rodrigues et al (2013, BMC Genomics), also found chitinase (expansin) and other genes (including PR-1 in time course of infection) involved in citrus responses to X. fastidiosa. Observe that in the line 211 the authors mentioned “ …it can be part of the cell-wall remodeling in response to the pathogen”, the similar hypothesis was already done by Lin et al (2007)….. “Such cell wall modifications have been hypothesized to be physical barriers to limit further pathogen invasion” and reinforced by Rodrigues et al (2013) that concluded “This work demonstrated that the defense response to the perception of bacteria involves cell wall modification…”. These are only some examples of many other statements done in the manuscript already published by other authors, but no mention of these previous studies was included in discussion.

2. Reading the previous studies, it seems that there is a good correlation about the hypothesis of host (grape and citrus) defense response against X. fastidiosa, among the data obtained using gene expression and protein expression showed in this work. However, in general, the authors mentioned the opposite (lines 266-275). I suggest the authors use the data already published for this pathogen to reinforce the data obtained in this manuscript. In this way CHURNER analysis could be shown as a power tool to improve the knowledge of the gene/protein for further investigation in functional approaches such as transgenic, breeding and others.
Minor:

1. Xf abbreviation in the manuscript should be replaced to X. fastidiosa.
2. The phrase in lines 52 – 54 needs a reference.
3. In MM section: Xylem sap collection and protein precipitation.
More information should be added in this section. Were those plants kept in field condition? The plants showed PD symptoms? If so, in which stage? How did you get 30-50 ml of xylem sap using only three plants? By drip of cut stem? Are you sure of this information? Please explain better this methodology, since high amount of xylem sap is very difficult to be obtained.
4. Results and Discussion. Lines 156 – 158. In table 1 is shown many genes that are located in the plant cell, such as RLKs (receptor like kinases). How to explain these proteins be found in xylem sap?
5. Lines 199 – 204. Note that similar observation, where modification of cell wall by lignification of xylem cells in response to X. fastidiosa were also recently showed by Niza et al., 2015 (Plant Pathology). In addition there other related work done by research groups led by Labavitch and Walker that could be also used to improve the discussion.
6. Line 310. Please change cream to ‘cream’.

---

## Round 0.2 · Minor Revisions

Please revise your manuscript concerning the remaining comments of reviewer 1.

Reviewer 1 ·

Basic reporting

I am satisfied with the authors major revisions and feel that this manuscript meets PeerJs standards for publication. I only suggest minor revisions for further clarification.

Line 70; Drop “THE” from “The Vitis vinifera cv. Chardonnay xylem sap…”

Line 74/75; The authors provide references “Yang and collaborators (2011) and a recent contribution by Katam et al. (2015)”, but there is little discussion on what those studies found. This sentence appears slightly underdeveloped. What did they see in those studies? Is it relevant?

Line 206; This makes it sound like your table 1 is of previously identified proteins. You should emphasize that these proteins were ALSO detected in your proteome. Please clarify this sentence

Line 268; Was this performed on plants used for exudate collection? If so, please indicate in text.

Line 331; do not need an apostrophe for LTPs.

TABLE 1 –Perhaps the Title of Table 1 should mention that these proteins were identified in xylem sap proteomes of infected and healthy plants.

Experimental design

Authors addressed all of my previous comments.

Validity of the findings

Authors addressed all of my previous comments.

Additional comments

Authors addressed all of my previous comments. I suggest only minor revisions.

Reviewer 2 ·

Basic reporting

This study adds interesting information regarding the analyses and interpretation of proteins function originated from proteomic data in plant-pathogen interaction studies.

Experimental design

No comments

Validity of the findings

No comments

Additional comments

The authors did a great job in response to reviewers' comments. I recommend the manuscript to be published in PeerJ Journal.

---

## Round 0.3 · accepted · Accept

This manuscript is now accepted after authors made necessary minor changes.